# C. elegans TFIIH subunit GTF-2H5/TTDA is a non-essential transcription factor indispensable for DNA repair

Karen L. Thijssen[1], Melanie van der Woude[1], Carlota Davó-Martínez[1], Dick H. W. Dekkers[2], Mariangela Sabatella [1,3], Jeroen A. A. Demmers [2], Wim Vermeulen [1] & Hannes Lans [1✉]

The 10-subunit TFIIH complex is vital to transcription and nucleotide excision repair. Hereditary mutations in its smallest subunit, TTDA/GTF2H5, cause a photosensitive form of the rare developmental disorder trichothiodystrophy. Some trichothiodystrophy features are thought to be caused by subtle transcription or gene expression defects. TTDA/GTF2H5 knockout mice are not viable, making it difficult to investigate TTDA/GTF2H5 in vivo function. Here we show that deficiency of C. elegans TTDA ortholog GTF-2H5 is, however, compatible with life, in contrast to depletion of other TFIIH subunits. GTF-2H5 promotes TFIIH stability in multiple tissues and is indispensable for nucleotide excision repair, in which it facilitates recruitment of TFIIH to DNA damage. Strikingly, when transcription is challenged, gtf-2H5 embryos die due to the intrinsic TFIIH fragility in absence of GTF-2H5. These results support the idea that TTDA/GTF2H5 mutations cause transcription impairment underlying trichothiodystrophy and establish C. elegans as model for studying pathogenesis of this disease.

[1] Department of Molecular Genetics, Oncode Institute, Erasmus MC Cancer Institute, Erasmus University Medical Center, 3015 CN Rotterdam, The Netherlands. [2] Proteomics Center, Erasmus University Medical Center, 3015 CN Rotterdam, The Netherlands. [3] Present address: Mariangela Sabatella, Princess Máxima Center for pediatric oncology, Heidelberglaan 25, 3584 CT Utrecht, The Netherlands. ✉email: w.lans@erasmusmc.nl

The general transcription factor TFIIH is an evolutionary conserved 10-subunit complex that has essential functions in transcription initiation and nucleotide excision repair (NER)[1]. It consists of a core complex comprised of subunits XPB/ERCC3, XPD/ERCC2, p62/GTF2H1, p44/GTF2H2, p34/GTF2H3, p52/GTF2H4 and p8/TTDA/GTF2H5 and the cyclin-activating kinase (CAK) complex consisting of MAT1/MNAT1, CDK7 and Cyclin H/CCNH[2]. In RNA polymerase II (Pol II)-driven transcription initiation, TFIIH binds the promoter and, stimulated by TFIIE[3,4], facilitates promoter escape and RNA synthesis by DNA helix opening using its XPB helicase/translocase subunit[5–7] and by phosphorylating the C-terminal domain of RPB1, the largest Pol II subunit, via its CDK7 subunit[8]. TFIIH is furthermore thought to function in RNA Pol I transcription[9,10] and to regulate the recruitment and/or activity of various transcriptional regulators[11,12]. In NER, TFIIH functions in both the global genome (GG-NER) and the transcription-coupled (TC-NER) subpathway. GG-NER repairs helix-distorting single-stranded DNA lesions anywhere in the genome, whereas TC-NER repairs any lesion that blocks elongating Pol II[13,14]. TFIIH is recruited to lesions upon DNA damage detection and, together with NER factor XPA, opens the DNA using its XPD helicase subunit, thereby verifying the presence of damage and providing a substrate for downstream endonucleases ERCC1/XPF and XPG to cut out the damaged DNA[15,16]. These essential functions of TFIIH are conserved from yeast to humans.

Mutations in the individual subunits of TFIIH cause several diseases characterized by growth and neurodevelopmental failure[17,18]. Mild mutations in XPB and XPD, which only affect TFIIH function in GG-NER, cause xeroderma pigmentosum, which is characterized by sun sensitivity and cancer susceptibility. More disruptive mutations in these subunits additionally cause Cockayne syndrome (CS) features, such as growth failure and progressive neurodegeneration, of which the exact pathogenesis is still debated and not fully understood. We proposed that specific impairment of TC-NER, in combination with prolonged lesion stalling of Pol II or the NER core complex, causes or contributes to CS features[19,20]. However, other causes such as specific transcriptional or mitochondrial defects have been put forward[21,22]. A third group of mutations in XPB and XPD, likely those affecting also TFIIH transcription function, cause a photosensitive form of trichothiodystrophy (TTD), which is furthermore characterized by brittle hair and nails, ichthyosis and progressive mental and physical retardation[23,24]. Besides XPB and XPD, only mutations in TTDA, the smallest subunit of TFIIH, have thus far been associated with photosensitive TTD[25]. Mutations in XPB, XPD and TTDA are rare and often found in compound heterozygous combinations. This, and the fact that so far only patients with mutations in three out of ten TFIIH subunits have been identified, reflects the fact that the TFIIH complex is essential for life[26–29].

In mice, disruption of XPD leads to embryonic lethality at the two-cell stage[27]. In contrast to other TFIIH subunits, yeast strains[30] and mouse embryonic stem cells and fibroblasts[28] with complete inactivation of TTDA are viable, suggesting that TTDA is not essential for TFIIH basal transcription function. However, also disruption of TTDA leads to embryonic lethality in mice, albeit that embryos survive almost up to birth[28]. Rare human TTD patients carry missense or nonsense mutations in TTDA that do not completely disrupt TTDA but lead to expression of a partially functional protein[28,31]. This suggests that although TTDA appears to be less essential for transcription than other TFIIH subunits, full TTDA loss is not compatible with development of multicellular life[28,32,33]. It is currently unclear why TTDA appears dispensable for transcription and viability of single cell systems but not of multicellular organisms and certain differentiated tissues.

The nematode C. elegans conserves many features of mammalian transcription and DNA repair[34,35]. We and others have previously shown that NER is conserved and protects genome integrity in germ cells and transcriptional integrity in somatic cells against DNA damage induced by UV light and other environmental and metabolic sources[36–41]. Thus far, the composition and activity of C. elegans TFIIH has not been addressed. Here, we focus specifically on the C. elegans TTDA ortholog GTF-2H5 and show that this TFIIH factor is dispensable for development and viability under normal, unchallenged laboratory culture conditions. However, GTF-2H5 is essential for TFIIH stability and genome maintenance via NER and becomes vital to transcription when this is challenged.

## Results and discussion

### C. elegans lacking TTDA/GTF-2H5 is viable.
To study the function of TTDA/GTF2H5 in C. elegans, we characterized animals carrying a deletion mutation (tm6360) in the orthologous gtf-2H5 gene. This gene only consists of two exons and is predicted to encode a 71 amino acid protein with an estimated molecular weight of 8.2 kD, which is similar to its human TTDA ortholog with which it shares 45% sequence identity. The tm6360 allele was obtained from the National Bioresource Project for the nematode[42] and represents a deletion of the entire second exon and flanking sequences (Fig. 1a). By RT-PCR and RT-qPCR of the mRNA, we confirmed that gtf-2H5 mutants do not express the second exon but found that the first exon is still expressed, albeit at >70% reduced levels (Fig. 1b, c; Supplementary Fig. 1a). To confirm whether indeed the tm6360 allele encodes a truncated protein, we determined by PCR and sequencing of cDNA whether a mutated mRNA was being produced. In the gtf-2H5 mutant, we detected by PCR a cDNA fragment stretching from the first exon of gtf-2H5 (primer 1 in Fig. 1a) to the last exon of the downstream B0353.1 gene (primer 4 in Fig. 1a; Supplementary Fig. 1b), which after sequencing revealed that gtf-2H5 mutants carrying the tm6360 allele express low levels of a mutant mRNA consisting of the first gtf-2H5 exon fused to the reverse sequence of the last exon of B0353.1 (Supplementary Fig. 1c). Gene prediction by FGENESH[43] indicated that this mRNA encodes a mutated GTF-2H5 protein of which the C-terminal half is deleted and replaced by nonsense amino acid sequence (Fig. 1d). Thus, tm6360 is likely a strong loss-of-function allele.

TFIIH consists of ten subunits that are all conserved in C. elegans (Table 1)[37] and which are all essential to life in mammals, including TTDA[26–29]. Strikingly, we found that gtf-2H5 mutants were viable, showed no embryonic lethality and produced similar brood size as wild type animals (Table 2). In human TTDA deficient patients, thus far one missense (L21P) and three truncating (M1T, E55X and R56X) mutations in TTDA have been identified[25,32], which are all thought to lead to expression of a partially functional TTDA protein that supports some TFIIH function and is thus still compatible with life[28,33]. Comparison of these mutations to the truncating tm6360 C. elegans mutation suggests that tm6360 does not similarly lead to expression of a partially functional protein, as GTF-2H5 is truncated in the middle of a region (from amino acids 16–55) that is retained in all human truncated, partially functional mutant TTDA proteins (Supplementary Fig. 1d). Still, to test whether the viability of gtf-2H5 mutants might be due to weak expression of partially functional truncated GTF-2H5, we examined whether full depletion of GTF-2H5 is lethal for C. elegans. To this end, we fused an auxin-inducible degradation (AID) tag[44] together with GFP to GTF-2H5 by knocking in both tags at the C-terminus of the gtf-2H5 gene using CRISPR/Cas9 (Fig. 2a). As comparison, we also knocked in AID and GFP at the N-terminus of gtf-2H1 (Fig. 2b), the C. elegans ortholog of TFIIH subunit GTF2H1/p62,

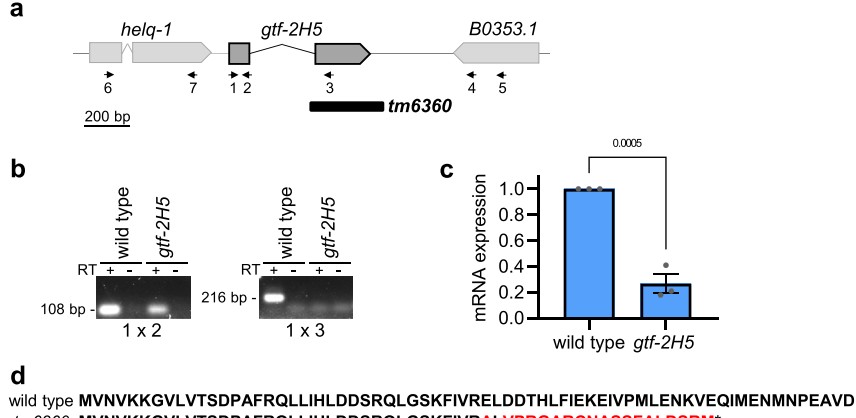

**wild type** MVNVKKGVLVTSDPAFRQLLIHLDDSRQLGSKFIVRELDDTHLFIEKEIVPMLENKVEQIMENMNPEAVDK*
**tm6360** MVNVKKGVLVTSDPAFRQLLIHLDDSRQLGSKFIVRALVPRGARCNASSFALDSRM*

**Fig. 1 The *gtf-2H5(tm6360)* allele encodes a truncated protein. a** Schematic depiction of *gtf-2H5* locus with parts of flanking genes *helq-1* and *B0353.1*. Numbered arrows indicate positions of primers used in RT-(q)PCR. Primer sequences are listed in Supplementary Table 2. The deletion of the *tm6360* allele is indicated with a black box. **b** PCR with primer pairs 1 × 2 or 1 × 3, as indicated in **a**, on cDNA of wild type or *gtf-2H5* animals. Lanes containing cDNA generated by reverse transcription of mRNA are labeled with '+'. As control, PCR on cDNA reaction samples without reverse transcriptase (RT) were included, labeled with '−'. Full image of the gel is shown in Supplementary Fig. 4. **c** Relative mRNA expression levels of exon 1 of the *gtf-2H5* transcript as determined by qPCR on wild type and *gtf-2H5* mutant animals. mRNA levels were normalized to wild type. Results are plotted as average with SEM (error bars) of three independent experiments. *p* value indicating statistical significance is shown. **d** Amino acid sequence of wild type GTF-2H5 and of the truncated protein predicted to be expressed by the *tm6360 gtf-2H5* allele. Red color indicate sequence that deviates from wild type. Numerical data are provided in Supplementary Data 2.

**Table 1 *C. elegans* TFIIH subunits.**

| *C. elegans* gene | Human ortholog |
| --- | --- |
| *xpb-1* | ERCC3 (XPB) |
| *xpd-1* | ERCC2 (XPD) |
| *gtf-2H1* | GTF2H1 (p62) |
| *gtf-2H2C* | GTF2H2 (p44) |
| *gtf-2H3* | GTF2H3 (p34) |
| *gtf-2H4* | GTF2H4 (p52) |
| *gtf-2H5* | GTF2H5 (p8) |
| *mnat-1* | MNAT1 (MAT1) |
| *cyh-1* | CCNH (Cyclin H) |
| *cdk-7* | CDK7 |

**Table 2 Brood size of wild type and *gtf-2H5* animals.**

| Strain | Brood size |
| --- | --- |
| | Average ± sem (*n*) |
| Wild type | 215 ± 14 (*n* = 20) |
| *gtf-2H5(tm6360)* | 212 ± 8 (*n* = 20) |

which in humans and yeast is essential for TFIIH integrity and function[2,45–47]. For simplicity, we will refer to the AID::GFP tag with 'AG' and to these knockin animals as *gtf-2H5::AG* and *AG::gtf-2H1*. Both knockin animals were fully viable and similarly expressed AG-tagged TFIIH in nuclei of all tissues (Fig. 2c; Supplementary Fig. 2). Also, both transgenic animals displayed normal, wild type UV survival in an assay that measures TC-NER (Supplementary Fig. 3a)[36,48], indicating that the TFIIH complex is intact and functional when either GTF-2H5 or GTF-2H1 is tagged. We crossed both strains with animals expressing Arabidopsis TIR1 (fused to mRuby) specifically in germ cells and early embryos under control of the *sun-1* promoter[44,49]. Exposure to auxin, which activates the TIR1 E3 ubiquitin ligase complex that ubiquitylates the AID tag, led to full depletion of both fusion proteins in the germline (Supplementary Fig. 3b). However, only the depletion of AID-tagged GTF-2H1, and

not that of GTF-2H5, caused complete embryonic lethality (Fig. 2d, e). Previously, we also showed that auxin-induced depletion of the TFIIH subunit XPB-1 causes developmental arrest[50]. As animals with truncated or depleted GTF-2H5 do not show any embryonic lethality, we conclude that unlike other TFIIH subunits, GTF-2H5 is not essential for embryogenesis and viability in *C. elegans*.

**GTF-2H5 promotes TFIIH stability in multiple tissues in vivo.** Both GTF-2H5 as well as GTF-2H1 showed exclusive nuclear expression in various tissues of *C. elegans* (Supplementary Fig. 2). To have an idea of the concentration of TFIIH molecules present in vivo in nuclei of different tissues, we determined GTF-2H5::AG and AG::GTF-2H1 concentrations in several readily visible tissues by carefully comparing average nuclear fluorescence signals to those of known GFP concentrations. This showed comparable concentrations of around 0.2 μM for GTF-2H1 in oocyte, hypodermal, intestinal and muscle nuclei (Fig. 3a), which is strikingly similar to concentration estimations made of Pol II in human fibroblasts (0.18 μM) using a similar method[51]. GTF-2H5 concentrations were mostly similar, but showed higher variation between individual nuclei and a lower concentration in muscle cells. This may reflect the notion that this factor can dynamically associate with TFIIH[52]. We quantified TFIIH concentration in brightly fluorescent cells that were easily discernable, but noticed that in many other nuclei, such as for instance in head neurons, expression levels seemed much lower (Supplementary Fig. 2). As nuclei in neurons are smaller than those in oocytes, hypodermal and intestinal cells, the TFIIH concentration will also be lower in these cell types. Similar analysis of mouse XPB-YFP fluorescence levels in tissues of XPB-YFP knockin mice also showed that TFIIH levels vary depending on the cell type, which, interestingly, was found to correlate to transcriptional activity in these cells[53]. Thus, possibly also in *C. elegans* TFIIH levels can be used as quantitative biomarker of transcriptional activity. This correlation, however, is difficult to prove as quantification of nascent mRNA by 5-ethynyl uridine labeling in different tissues of *C. elegans* does not produce sufficiently reproducible results in our hands[50].

To determine the impact of GTF-2H5 loss on TFIIH levels, we crossed *AG::gft-2H1* animals with *gtf-2H5* mutants and found that loss of GTF-2H5 led to reduced levels of AG::GTF2H1 in all tested

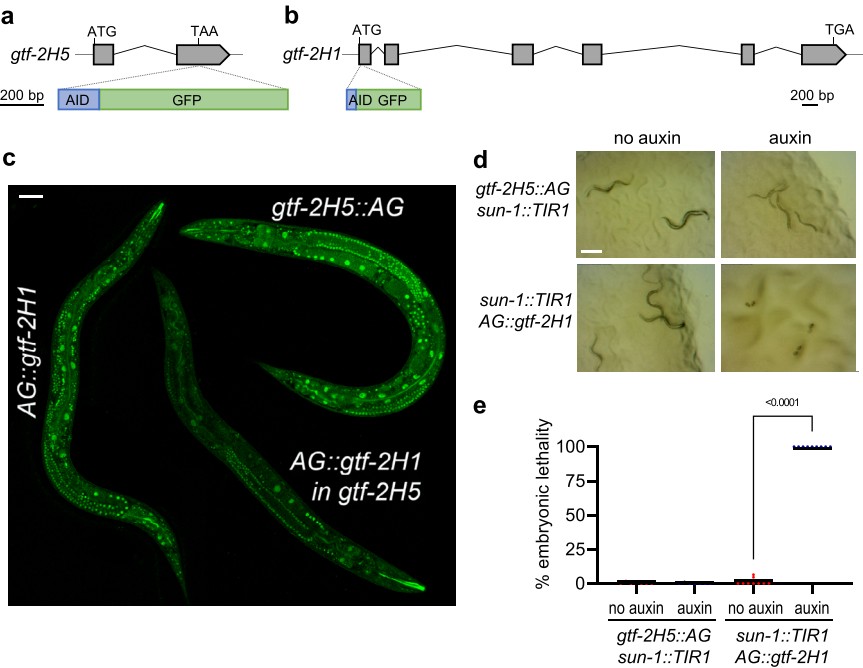

**Fig. 2 GTF-2H5 and GTF-2H1 are ubiquitously expressed but at different concentrations. a** Schematic depiction of the *gtf-2H5* locus and C-terminal knockin site of the AID::GFP tag. **b** Schematic depiction of the *gtf-2H1* locus and N-terminal knockin site of the AID::GFP tag. **c** Composite overview image generated by merging independent confocal scans of *gtf-2H5::AG, AG::gtf-2H1* and *gtf-2H5; AG::gtf-2H1*. Scale bar: 50 μm. **d** Stereo microscope view of the offspring of *gtf-2H5::AG* and *AG::gtf-2H1* animals expressing TIR1 under control of the *sun-1* promoter grown in absence or presence of 1 mM auxin. Only auxin-induced depletion of *AG::gtf-2H1* leads to embryonic lethality. Scale bar: 0.2 mm. **e** Quantification of the experiment described in **d**. Shown is a scatter dot plot of the average percentage embryonic lethality observed on at least seven plates for each condition in two independent experiments. *p* value indicating statistical significance is shown. Numerical data are provided in Supplementary Data 2.

tissues by more than 50% (Fig. 3a). These results are in line with lowered TFIIH protein levels observed in patient and mouse fibroblasts[28,54], while TFIIH subunit mRNA levels are unaffected[55], and indicate that GTF-2H5 has an evolutionary conserved function in maintaining steady-state levels of TFIIH, not only in vitro but also in various tissues in vivo. To determine whether GTF-2H1 levels are lower because the TFIIH complex itself is not intact in *gtf-2H5* mutants, we performed immunoprecipitation (IP) using GFP as bait on AG::GTF-2H1 expressed in wild type and *gtf-2H5* animals, followed by mass spectrometry analysis. In wild type animals, the entire TFIIH complex co-immunoprecipitated with AG::GTF-2H1, including the small GTF-2H5 subunit of which three peptides were identified (Fig. 3b; supplementary Data 1). Also, many NER and transcriptional proteins known to have affinity for or associate indirectly with TFIIH were identified, such as XPA-1 and XPF-1. In *gtf-2H5* animals, we identified nine TFIIH subunits but not GTF-2H5, while the overall abundance of TFIIH subunits in the IP samples was lower (supplementary Data 1). These results confirm the absence of functional GTF-2H5 in these mutants as they indicate that the *tm6360* mutant protein, if it is at all expressed, is not part of the TFIIH complex. Furthermore, these results are in line with lowered TFIIH levels in *gtf-2H5* mutants, but indicate that the TFIIH complex itself, apart from GTF-2H5, is intact. Interestingly, also XPA-1 did not co-immunoprecipitate well with AG::GTF-2H1 in *gtf-2H5* mutants, which may point to specific defects in TFIIH function in NER.

**GTF-2H5 is essential for nucleotide excision repair.** Because of the lowered TFIIH levels, we tested whether specific functions of TFIIH were affected and first focused on its role in NER. We and others have previously shown that UV-induced DNA damage causes embryonic lethality in the absence of GG-NER[36,38,39,48].

We observed that *gtf-2H5* mutants are as sensitive to UV-induced DNA damage as NER-deficient *xpa-1* mutants, showing severe embryonic lethality upon UV irradiation (Fig. 4a). These results further confirm that the *tm6360* allele is a null allele and does not encode partially functional GTF-2H5.

To gain further insight into the cause of UV hypersensitivity of *gtf-2H5* mutant animals, we visualized recruitment of TFIIH to DNA damage. In *C. elegans* oocytes, DNA is condensed and organized into six pairs of bivalents, i.e., paired homologous chromosomes, that are readily discernable by microscopy e.g., when stained with DAPI (Supplementary Fig. 3c, left panel). We showed previously that upon UV irradiation, the NER endonuclease ERCC-1/XPF-1 rapidly re-localizes from the nucleoplasm to damaged chromosomes, reflecting its activity in NER[50]. This damaged-DNA binding is dependent on GG-NER and lasts for approximately half an hour until UV photolesions are repaired and NER is completed. In line with this, we observed clear re-localization of both GTF-2H5::AG and AG::GTF-2H1 to oocyte bivalents 10 min after UV irradiation, while imaging living (Fig. 4b) or fixed (Supplementary Fig. 3c) animals, reflecting TFIIH binding to UV-damaged DNA. This recruitment was not observed anymore 35 min after UV, indicating completion of repair, as previously shown[50]. Next, we tested re-localization of AG::GTF-2H1 in *gtf-2H5* mutants and strikingly observed that its UV-induced recruitment to damaged chromosomes was strongly reduced after 10 min and persisted after 35 min (Fig. 4b). We observed similar *gtf-2H5*-dependent recruitment to damaged bivalents of another TFIIH subunit, XPB-1 (Supplementary Fig. 3d), of which we previously had generated AID::GFP knockin animals[50]. Together with the strong UV hypersensitivity of *gtf-2H5* mutants, these results indicate that GTF-2H5 is needed for efficient binding of TFIIH to damaged DNA and that in absence of GTF-2H5 repair cannot be completed. Also in vitro

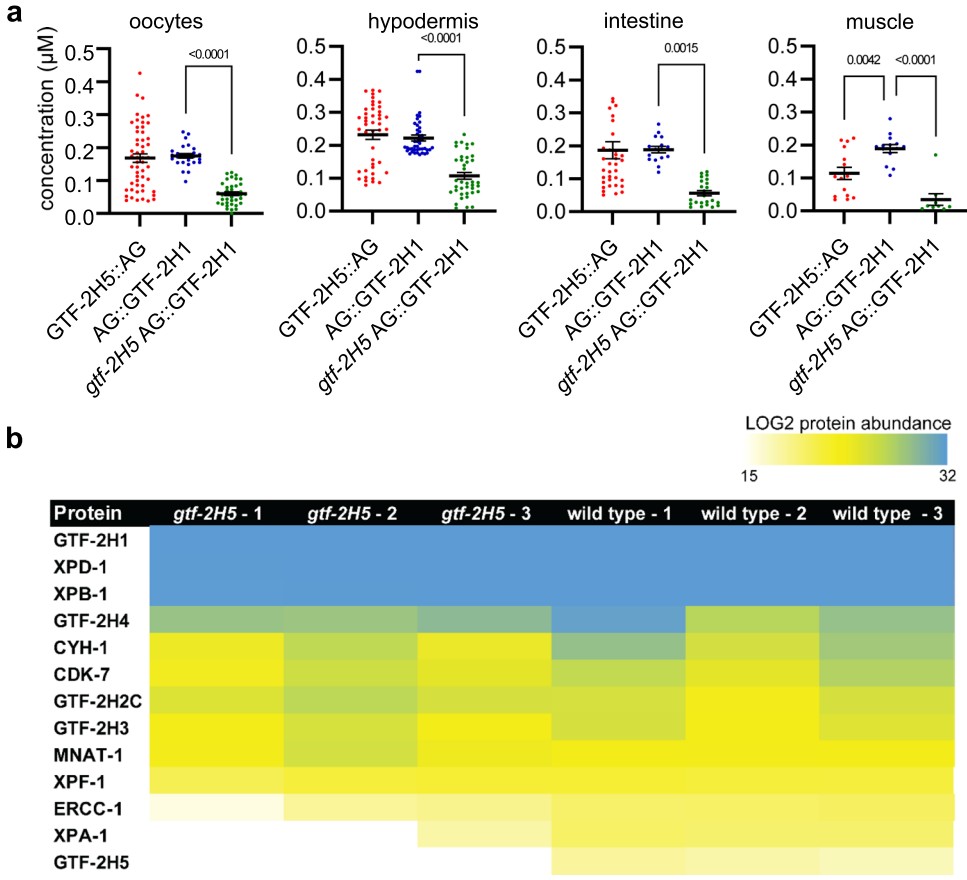

**Fig. 3 TFIIH protein levels and composition in *gtf-2H5* mutants. a** Quantification of GTF-2H5::AG and AG::GTF-2H1 concentration in nuclei of oocytes, hypodermal, intestinal and muscles cells of wild type and *gtf-2H5* animals. Concentration was determined by comparison of the average fluorescence levels in the entire nucleus to the fluorescence of known concentrations of purified GFP. *p* values indicating statistical significance are shown. Numerical data are provided in Supplementary Data 2. **b** Heat map representation of protein abundance, based on summed peptide intensities (normalized to bait AG::GTF-2H1), of TFIIH subunits and NER proteins XPF-1, ERCC-1 and XPA-1 in three replicate AG::GTF-2H1 immunoprecipitation experiments analyzed by mass spectrometry.

and in cultured mammalian cells TTDA was found to be essential for NER and the recruitment of TFIIH to sites of UV-induced DNA damage, likely by stimulating XPB ATPase activity and TFIIH translocase activity together with XPC and XPA[25,28,56,57]. Our results show that GTF-2H5 has a similar, evolutionary conserved role in stimulating TFIIH activity in NER in oocytes of *C. elegans*. Besides NER, TTDA has been implicated in repair of oxidative DNA damage, as TTDA knockout mouse embryonic stem cells are sensitive to ionizing radiation and $KBrO_3$[28]. However, when comparing $KBrO_3$ sensitivity of *C. elegans gtf-2H5* mutants and wild type animals, we did not observe any obvious hypersensitivity (Fig. 4c).

Since *gtf-2H5* mutants are as UV hypersensitive as *xpa-1* mutants, we tested whether these animals also exhibit phenotypes that we previously reported to be caused by accumulating DNA damage and mutations due to NER deficiency. A significant proportion of *C. elegans* NER-deficient animals in a population exhibit growth delay due to spontaneous DNA damage[58]. Indeed, we observed that a percentage of *gtf-2H5* animals exhibited delayed development, just as *xpa-1* animals, in comparison to wild type (Fig. 4d). Also, we previously showed that NER mutants do not have reduced postmitotic lifespan, i.e., longevity measured from the onset of adulthood, but do show severely reduced replicative lifespan, i.e., transgenerational replicative capacity[58]. We confirmed again that *xpa-1* mutants have a postmitotic lifespan comparable to wild type animals (Fig. 4e), but that their ability

to proliferate and reproduce over generations strongly declines (Fig. 4f). To our surprise, however, we found that *gtf-2H5* mutants show both a shortened postmitotic (Fig. 4e) and replicative (Fig. 4f) lifespan, which appear both more reduced than of *xpa-1* mutants. Thus, besides being clearly NER defective, *gtf-2H5* animals likely carry a defect in an additional biologic pathway that causes them to live shorter.

**GTF-2H5 is required for transcription when this is challenged.** TTD patients with mutations in TTDA, XPB and XPD exhibit typical TTD features like brittle hair and nails and ichthyosis and, in addition, photosensitivity and progressive neuropathy that are reminiscent of xeroderma pigmentosum[23,24]. Photosensitivity and part of the neuropathy are considered to be due to defects in NER, but the other features are thought to be caused by limited availability of TFIIH, due to its instability and lowered levels that lead to its exhaustion in terminally differentiated cells, such as skin and hair keratinocytes, before the transcriptional program in these cells has ended[59,60]. As TFIIH is needed for transcription of abundant proteins in these cells, such as cysteine-rich matrix proteins, reduced amounts of TFIIH likely explain the sulfur-deficient brittle hair and nails and scaly skin features. Thus, we postulate that, similarly, the lowered TFIIH levels in *gtf-2H5* animals lead to TFIIH exhaustion and limited TFIIH availability for transcription initiation in older animals, causing cell dysfunction and reduced lifespan.

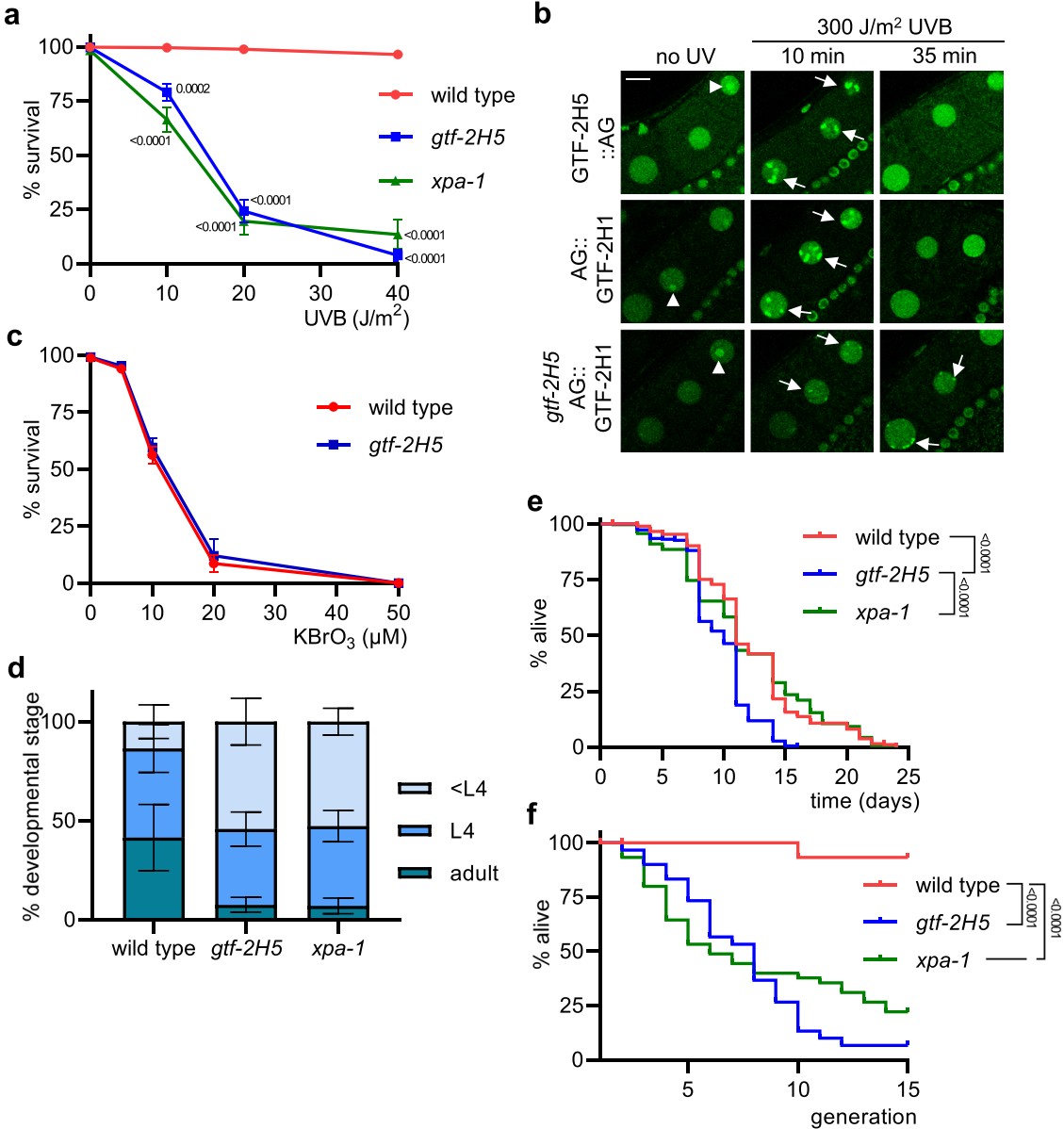

**Fig. 4 *gtf-2H5* animals are NER deficient and show diminished growth and lifespan. a** Germ cell and embryo survival assay after UVB irradiation of germ cells in young adult wild type, *gtf-2H5* and *xpa-1* animals. The percentages of hatched eggs (survival) after UVB irradiation are plotted against the applied UV-B doses. Results are plotted as average with SEM (error bars) of at least seven experiments. Number of animals counted, respectively for UV dose 0, 10, 20, 30 and 40 J/m², for wild type $n = 735, 820, 832$ and 738; for *gtf-2H5* $n = 725, 495, 314$ and 117; for *xpa-1* $n = 480, 284, 156$ and 83. $p$ values indicating statistical significant difference compared to wild type are shown. **b** Representative images showing real-time recruitment of GTF-2H5::AG or AG::GTF-2H1 to UVB-damaged chromosomes in oocytes of living wild type (upper and middle panel) or *gtf-2H5* (lower panel) animals, before UVB irradiation (no UV) and 10 min and 35 min after 300 J/m² UVB irradiation. Recruitment to paired homologous chromosomes are indicated with arrows. Nucleoli are indicated with an arrowhead. Scale bar: 10 μm. **c** Survival assay after incubation of wild type and *gtf-2H5* L1/L2 larvae with increasing concentrations of KBrO₃ for 24 h, which induces oxidative DNA damage. The percentages of non-arrested, developing larvae (survival) are plotted against the applied KBrO₃ concentration. Results are plotted as average with SEM (error bars) of two independent experiments each performed in triplicate. Number of animals counted, respectively for KBrO₃ dose 0, 5, 10, 20, and 50 μM, for wild type $n = 637, 756, 707, 558$ and 456; for *gtf-2H5* $n = 708, 688, 894, 505$ and 467. **d** Quantification of larval growth of wild type, *gtf-2H5* and *xpa-1* animals by determining the percentage of adult, L4 and younger than L4 (<L4) animals observed 48 h after animals are laid as eggs at 25 °C. Results are plotted as average with SEM (error bars) of at least four independent experiments. Number of animals counted are for wild type $n = 905$, for *gtf-2H5* $n = 846$ and for *xpa-1* $n = 456$. **e** Post-mitotic lifespan analysis showing the percentage of alive adult wild type ($n = 290$), *gtf-2H5* ($n = 300$) and *xpa-1* ($n = 285$) animals per day. $p$ values indicating statistical significance are shown. **f** Replicative lifespan analysis showing the percentage survival of successive generations of wild type, *gtf-2H5* and *xpa-1* animals if, in each generation, one animal is passaged. Depicted are cumulative results from at least two independent experiments ($n = 15$ per experiment). $p$ values indicating statistical significance are shown. Numerical data are provided in Supplementary Data 2.

To test whether the lowered TFIIH amounts in *gtf-2H5* animals indeed become restrictive in circumstances with altered transcriptional demand, we tested whether a transcription defect might become apparent in these animals when transcription is challenged. As transcription cannot be reliably measured with 5-ethynyl uridine labeling, we crossed *gtf-2H5* mutants with animals expressing GFP in body muscle cells under control of the *eft-3* promoter[44], and quantified GFP fluorescence levels as proxy for transcriptional competence[50]. This showed no difference between wild type and *gtf-2H5* animals (Fig. 5a, b), indicating that in unperturbed conditions *gtf-2H5* animals are fully transcriptionally competent for this muscle-specific transgene.

Next, we compromised transcription by limiting the availability of another transcription initiation factor, without compromising NER. To this end, we cultured wild type and *gtf-2H5* animals on *gtf-2E1* RNAi bacteria to diminish protein levels of the essential TFIIE transcription factor complex, which stimulates TFIIH activity and helps anchor it within the Pol II pre-incision complex, but has no role in NER[3,4,61–64]. *gtf-2E1* RNAi food is not fully efficient and causes only partial knockdown of *gtf-2E1*

transcripts and therefore only mild embryonic lethality in wild type animals (Fig. 5c, d). Strikingly, however, *gtf-2H5* animals grown on the same *gtf-2E1* RNAi food showed very high levels of embryonic lethality. By measuring GFP fluorescence in body wall muscles, we observed that partial *gtf-2E1* knockdown lowered GFP proteins levels in wild type animals, which was further enhanced in *gtf-2H5* animals (Fig. 5a, e). Thus, in conditions of limited availability of the TFIIE transcription factor and lowered gene expression, GTF-2H5 becomes essential for transcription efficiency and viability. To confirm this in another way, we measured the transcript levels of two housekeeping genes, *cdc-42* and *pmp-3*, by qPCR in cDNA generated from RNA isolated from the same number of non-gravid adult wild type or *gtf-2H5* animals grown on control or *gtf-2E1* RNAi. This showed that *gft-2H5* knockout or *gtf2E1* knockdown alone did not result in lowered transcript levels, but the combined loss of *gtf-2H5* and *gtf-2E1* strongly reduced transcription of both housekeeping genes (Fig. 5f, g). These results therefore suggest that in *C. elegans* GTF-2H5 (and thus high steady-state TFIIH levels) is dispensable for transcription throughout most of *C. elegans* lifespan in

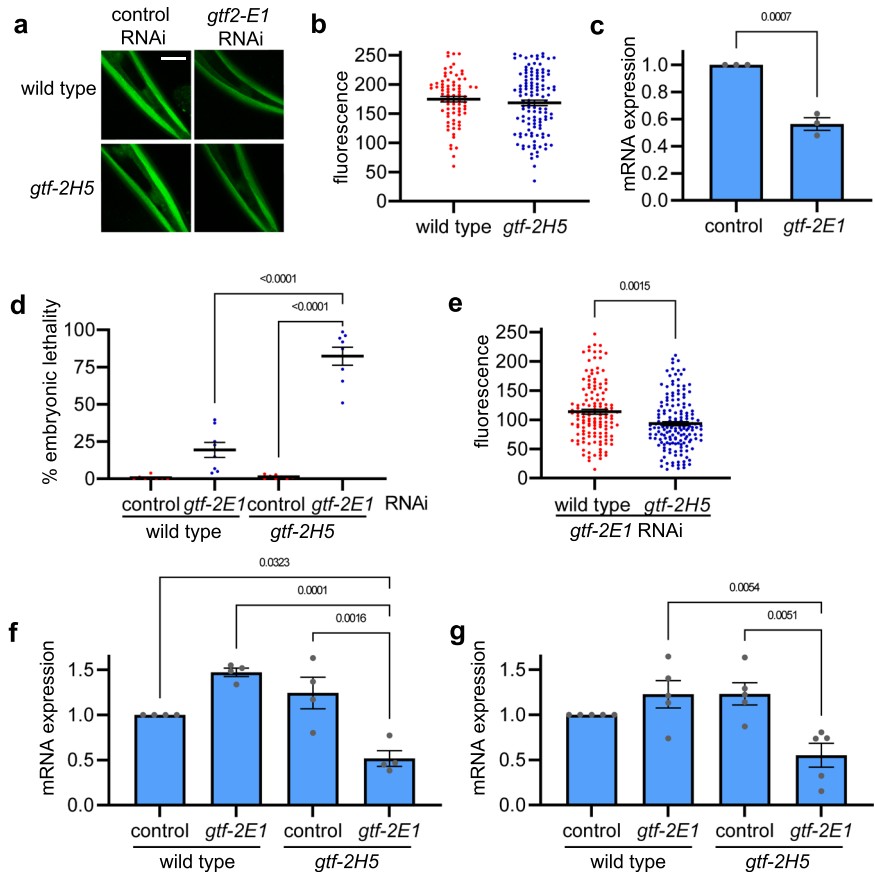

**Fig. 5 GTF-2H5 promotes transcription when this is challenged. a** Representative images of wild type and *gtf-2H5* animals expressing AID::GFP under control of *eft-3* promoter in body wall muscles, shown here in the head of *C. elegans*, grown on control or *gtf-2E1* RNAi. Scale bar: 25 μm. **b** Scatter dot plot showing average and SEM of the relative GFP fluorescence levels in head muscle cells of wild-type and *gtf-2H5* animals, as depicted in **a**. **c** Relative *gtf-2E1* mRNA levels as determined by qPCR on animals grown on control or *gtf-2E1* RNAi. Results are normalized to control RNAi and plotted as average with SEM (error bars) of three independent experiments. **d** Scatter dot plot showing the average percentage with SEM of embryonic lethality observed on eight plates in two independent experiments with either wild type animals or *gtf-2H5* animals (both also carrying the *eft-3::GFP* transgene used in **a** and **b**, grown on control or *gtf-2E1* RNAi food. Number of animals counted for wild type are n = 617 (control RNAi), n = 570 (*gtf-2E1* RNAi) and for *gtf-2H5* n = 597 (control RNAi), n = 419 (*gtf-2E1* RNAi). **e** Scatter dot plot showing average and SEM of the relative GFP fluorescence levels in head muscle cells of wild-type and *gtf-2H5* animals grown on *gtf-2E1* RNAi food, as depicted in **a**. **f** Relative *cdc-42* mRNA levels as determined by qPCR on wild type or *gtf-2H5* animals grown on control or *gtf-2E1* RNAi. Results are normalized to wild type on control RNAi and plotted as average with SEM (error bars) of four independent experiments. **g** Relative *pmp-3* mRNA levels as determined by qPCR on wild type or *gtf-2H5* animals grown on control or *gtf-2E1* RNAi. Results are normalized to wild type on control RNAi and plotted as average with SEM (error bars) of five independent experiments. *p* values indicating statistical significance are shown. Numerical data are provided in Supplementary Data 2.

unperturbed laboratory culturing conditions, but becomes essential for transcription if this process is somehow compromised or challenged.

This notion may explain contradicting results in previous studies of TTDA's importance to transcription. Initially, upon its identification, human TTDA was considered a NER-specific TFIIH factor dispensable for transcription. TTD-A patient cells had no obvious transcriptional defects and purified TTDA did not stimulate TFIIH-dependent transcription in vitro[52,55,56,65]. However, TTDA knockout mouse embryonic stem cells did show reduced transcription levels[28] and a reconstituted transcription assay using different recombinant TFIIH also showed that transcription was stimulated by TTDA[66]. Furthermore, yeast lacking TTDA ortholog TFB5 did not show any major change in gene expression as observed by microarray, but still TFB5 was found to stimulate transcription in vitro and efficient transcription in vivo when there was high demand such as in changing environmental conditions[30]. Thus, it appears that depending on the particular conditions of the assay or cell type used, contingent on transcriptional demand, the absence of TTDA/GTF-2H5 may or may not be a limiting factor. This feature may explain why cultured yeast and mammalian cells and C. elegans can thrive without TTDA/GTF-2H5, whereas mice with full TTDA loss are not viable, likely because mammalian embryogenesis requires high transcriptional capacity[60].

High resolution cryo-EM structures of the human preinitiation complex do not suggest a direct role for TTDA in loading or DNA interaction of TFIIH at promoters[67]. Therefore, to test whether the importance of GTF-2H5 for compromised transcription is due to its ability of stabilizing TFIIH, we tested if lowering

TFIIH levels in another way, by partially depleting the TFIIH subunit GTF-2H2C (p44/GTF2H2 in humans; Table 1), led to similar enhanced embryonic lethality when transcription is compromised. To this end, we cultured wild type animals on an equal mixture of control RNAi and gtf-2H2C RNAi food, which destabilized TFIIH, as visualized by lowered AG::GTF-2H1 levels (Fig. 6a), and caused incompletely penetrant embryonic lethality (Fig. 6b). However, when gtf-2H2C RNAi food was equally mixed with gtf-2E1 RNAi food, this led to very high levels of embryonic lethality. These results confirm that unstable and lowered levels of TFIIH compromise transcription when this is challenged.

**gtf-2H5 C. elegans mutants as model for trichothiodystrophy.** TFIIH is an essential transcription and DNA repair factor. While thus far all TFIIH subunits were thought to be essential to multicellular life, including TTDA/GTF2H5, here we show that C. elegans strains with non-functional GTF-2H5, either by truncation due to the tm6360 allele or by depletion due to the AID degron tag, are viable. As gtf-2H5 mutants are strongly UV-hypersensitive, similar as xpa-1 loss-of-function mutants, and show impaired recruitment of TFIIH to UV-induced DNA damage, GTF-2H5 is an essential NER factor (Fig. 6c). With the characterization of GTF-2H5 presented in this study, and with it of the entire TFIIH complex, we and others have shown full functional conservation of both the GG- and TC-NER pathway in C. elegans[36,37,39,41,50,58,68–74].

In recent years, besides mutations in XPB, XPD and TTDA, multiple additional mutations in genes have been identified to

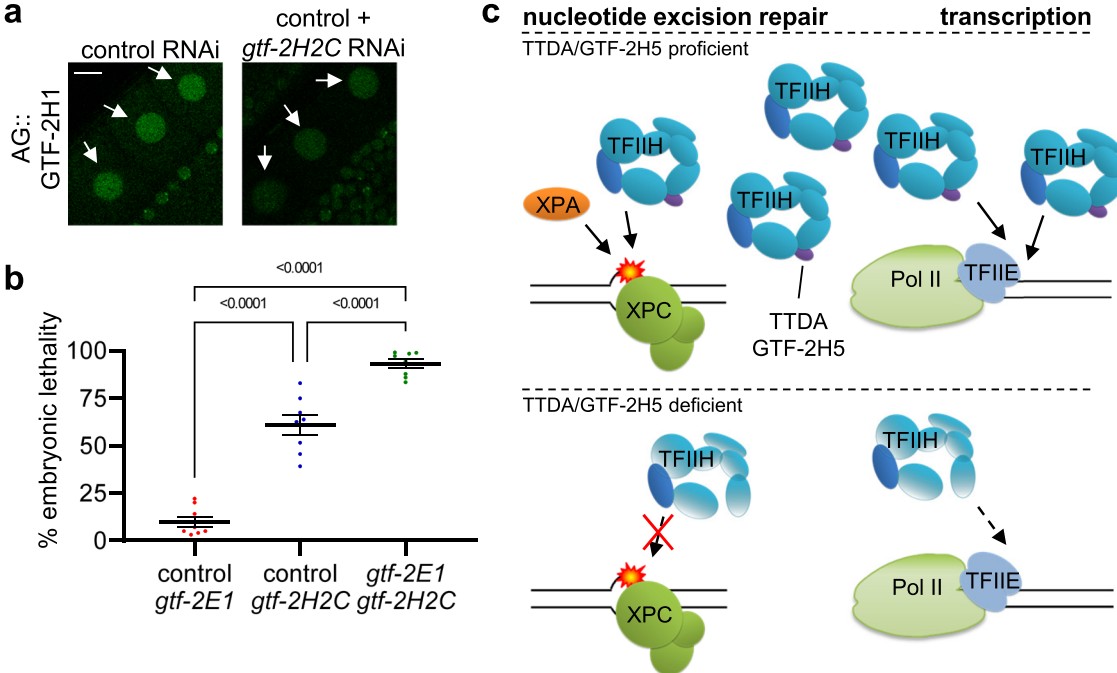

**Fig. 6 Synthetic lethality between GTF-2H2C and GTF-2E1 and model for GTF-2H5 activity. a** Representative images of AG::GTF-2H1 fluorescence in oocyte nuclei (indicated by arrows) of *AG::gtf-2H1* knockin animals grown on control RNAi or a 1:1 mixture of control and *gtf-2H2C* RNAi food. Scale bar: 10 μm. **b** Scatter dot plot showing the average percentage with SEM of embryonic lethality observed on eight plates in two independent experiments with wild type animals grown on 1:1 mixtures of control RNAi with *gtf-2E1* RNAi food (*n* = 574), control RNAi with *gtf-2H2C* RNAi food (*n* = 1009) or *gtf-2E1* RNAi and *gtf-2H2C* RNAi food (*n* = 730). *p* value indicating statistical significance is shown. Numerical data are provided in Supplementary Data 2. **c** Model for TTDA/GTF-2H5 involvement in nucleotide excision repair and transcription initiation. In wild type cells (upper part), TTDA/GTF-2H5 is the smallest subunit of a fully functional TFIIH complex, which exists in sufficiently high concentrations to promote nucleotide excision repair together with XPC and XPA and transcription initiation by RNA polymerase II (Pol II) together with TFIIE (and other not depicted repair and transcription factors). In TTDA/GTF-2H5 deficient cells (lower part), the TFIIH complex is not efficiently recruited and active in nucleotide excision repair. Also, as the complex is unstable and exists in low concentrations, it can only support transcription initiation if this is not too demanding.

cause TTD or TTD-like syndromes, among which are mutations in transcription initiation factor TFIIE, which all affect the stability of proteins involved in different steps of transcription and translation[62,63,75–79]. Together, these have put forward the unifying idea that TTD is caused by instability of proteins involved in gene expression. Our results are in line with this idea and confirm that loss of GTF-2H5 leads to instability of the transcription initiation factor TFIIH, which may lead to transcription problems in cells of aged animals due to TFIIH exhaustion (Fig. 6c). Furthermore, we find that when transcription initiation is hampered, by lowering of TFIIE levels, the reduced TFIIH levels in gtf-2H5 mutants become restrictive and cause strong embryonic lethality. Thus, in contrast to its function in NER, TTDA appears mostly dispensable for TFIIH function in facilitating transcription initiation by RNA polymerase II. However, in conditions of transcription stress, when there is demand for high TFIIH concentrations due to e.g high transcriptional output or reduced availability of other transcription initiation factors, the TFIIH-stabilizing ability of TTDA becomes essential to ensure sufficiently high TFIIH levels for transcription to initiate. As TFIIH promotes transcription elongation by RNA pol I as well, which was found to be impaired in TTDA deficient mouse embryonic stem cells[10,80], it would be interesting to investigate whether, similarly, TFIIH can become limiting for RNA pol I transcription in gtf-2H5 mutants. Taken together, our findings indicate that gtf-2H5 mutant C. elegans can be an advantageous multicellular animal model for studying genetic and developmental pathogenic aspects of TTD features.

## Material and methods

**C. elegans strains and culture.** C. elegans strains used are listed in Supplementary Table 1. C. elegans was cultured according to standard methods on nematode growth media (NGM) agar plates seeded with Escherichia coli OP50. All mutants were backcrossed against wild type strain, which was Bristol N2. gtf-2H5::AG knockin animals were generated using gRNA with targeting sequence CATGGAAAATATGAATCCGG and as homology directed repair template a heteroduplex DNA fragment generated by mixing PCR products[81] obtained with primer combinations 5′-GAACTCGACGATACGCATTTG-3′/5′-GCCTAAAACATG AAGCCTGTTG-3′ and 5′-GTATCCTTCAATCCACCGTTC-3′/ 5′-TTTGTATAGTTCGTCCATGCC-3′ on a gene fragment consisting of AID::GFP sequences flanked by 200 bp homology arms from the gtf-2H5 locus. AG::gtf-2H1 knockin animals were generated using gRNA with targeting sequence TTTTCA-GATGTCAGACGAGT and as homology directed repair template a gene fragment consisting of AID::GFP sequences flanked by 100 bp homology arms from the gtf-2H1 locus in pUCIDT-KANA (Integrated DNA technologies). sgRNA and plasmids were injected in the Cas9dPiRNA expressing strain HCL67 (a kind gift from Heng-Chi Lee)[82]. Knockin animals were verified by genotyping PCR and sequencing, after which the Cas9 was removed by backcrossing against wild type. RNAi bacteria were obtained from the Caenorhabditis elegans RNAi feeding library[83]. Control RNAi was vector pPD129.36 (a gift from Andrew Fire).

**Brood size, growth and embryonic lethality.** To measure brood size, L4 animals were individually seeded and transferred to a new culture plate every day, after which eggs laid were counted. To measure growth, adult animals were allowed to lay eggs for 4 h. After 48 h at 25 °C, the growth of the progeny was scored by counting the amount of adult, L4 and younger than L4 animals. To measure embryonic lethality of AID-tagged transgenic strains, late L4 animals were mock treated or exposed to 1 mM auxin (3-indoleacetic acid, Sigma) for 24 h. To measure embryonic lethality

of RNAi-fed strains, animals were first grown for one generation on RNAi food. Next, four animals per plate were allowed to lay eggs for 3 h, after which hatched and unhatched eggs were scored.

**Lifespan assays.** To measure post-mitotic lifespan, standard longevity assays were performed at 20 °C as previously described with day 1 defined as the day when animals reached adulthood[84]. Animals were scored every 1–3 days. To measure transgenerational replicative lifespan, assays were performed as previously described[58]. In brief, 15 animals per strain were individually seeded and cultured to produce progeny at 25 °C. From each progeny, a single animal was randomly picked, transferred to a fresh culture plate and grown to produce new offspring, which was repeated for 15 generations. Progeny was considered to have lost viability if animals arrested development, produced no or inviable progeny or died before reproduction. Animals that crawled off the plate were censored.

**RT-PCR and RT-qPCR.** For RT-PCR, animals were lysed in TRIzol (Qiagen) and RNA was purified using RNeasy spin columns (Qiagen). cDNA was generated using Superscript II Reverse Transcriptase (Invitrogen) according to manufacturer's instructions. For RT-qPCR, RNA was isolated and cDNA generated from 10 young adult animals per strain and condition using the Power SYBR Green Cells-to-Ct kit (Invitrogen) according to the method described by Chauve et al.[85]. qPCR was performed using PowerUp SYBR Green Master Mix (Invitrogen), according to manufacturer's instructions, with 58 °C as annealing temperature and 1 min elongation time. Primers used for RT-PCR and RT-qPCR are listed in Supplementary Table 2. cdc-42 and pmp-3 cDNA were used as reference genes, unless otherwise stated.

**Imaging and GTF-2H5 and GTF-2H1 concentration measurements.** For microscopy of living animals, animals were mounted on 2% agar pads in M9 containing 10 mM $NaN_3$ (Sigma) and imaged on a Zeiss LSM700 or Leica SP8 confocal microscope. For microscopy of fixed animals, animals were fixed on Poly-L-lysine hydrobromide (Sigma) slides with 4% paraformaldehyde in PBS and slides were mounted using Vectashield with DAPI (Vector laboratories). Fluorescence intensities of AID::GFP driven by eft-3 promoter in head body wall muscles were imaged and quantified using ImageJ software. To estimate concentrations of GTF-2H5::AG and AG::GTF-2H1, we imaged GFP fluorescence intensity of Z-slices of individual nuclei of the indicated cell-types. Average fluorescence intensities of each nucleus were then calibrated to fluorescence intensities of different concentrations of purified eGFP (Biovision), using similar imaging condition, to derive the protein concentration.

**Survival assays.** UV survival was carried out as described[36,48]. For UV irradiation, Philips TL-12 UV-B tubes (40 W) were used. To determine UV-induced embryonic lethality in Fig. 4a, staged young adult worms were washed, UVB-irradiated on agar plates without food and allowed to recover for 24 h on plates with OP50 bacteria. Next, five adults were allowed to lay eggs for 3 h on 6 cm plates seeded with HT115 bacteria, in five-fold for each UVB dose and 24 h later, the amount of hatched and unhatched (dead) eggs was counted. To determine UV-induced larval growth arrest in Supplementary Fig. 3a, eggs were collected by hypochlorite treatment of adult animals and plated onto agar plates with HT115 bacteria. After ~16 h, L1 larvae were UVB-irradiated and allowed to recover for 48 h. Survival was determined by counting animals arrested at the L1/L2 stages and animals that developed beyond the L2 stage. To determine $KBrO_3$ sensitivity, eggs were collected by hypochlorite treatment and allowed to hatch in S basal medium containing OP50 bacteria for 16 h at 21 °C while

shaking. Next, the indicated KBrO$_3$ concentrations were added to the liquid cultures, in triplicate. After 24 h incubation, animals were plated on 6 cm culture plates seeded with HT115 bacteria and incubated at 15 °C. After a recovery period of 72 h, the number of arrested animals at the L1/L2 stages and animals that developed beyond the L2 stage were counted. Survival percentages was calculated by dividing the number of arrested larvae by the total amount of animals.

**Immunoprecipitation and mass spectrometry**. For immuno-precipitation, animals from ten full 9 cm culture plates were added to a 500 ml liquid culture of S basal medium containing OP50 bacteria and grown for 5 days at 21 °C while shaking. Next, 500 ml ice cold M9 buffer was added and animals were collected in 100 ml by gravity sedimentation using a separation funnel. Animals were centrifuged for 3 min at $600 \times g$ with slow deceleration and washed by repeating this three times in ice cold M9. Next, the worm pellet was washed in 50 ml cold lysis buffer (50 mM HEPES pH 7.5, 1 mM EGTA, 1 mM MgCl2, 100 mM KCl, 10% glycerol, 0.05% NP-40 and complete EDTA free protease inhibitor) and frozen in liquid nitrogen. The frozen worms were lysed in IP buffer (30 mM HEPES pH 7.5, 130 mM NaCl, 1 mM MgCl2, 0.5% Triton X-100 and complete EDTA free protease inhibitor) for 10 min at 4 °C and sonicated using a Biorupter Sonicator (Diagenode). Sonicated samples were incubated with 500 U Benzonase (Sigma) for 1 h on ice. Lysates were cleared by centrifugation and supernatants were subjected to immunoprecipitation using GFP-TRAP-A agarose beads (Chromotec) for 2 h at 4 °C. Beads were collected by centrifugation and washed five times with IP buffer. Proteins were eluted from the beads by incubation at 95 °C for 5 min in 2x laemmli SDS sample buffer and separated on SDS-PAGE gel.

Each SDS-PAGE gel lane was cut into 1-mm slices using an automatic gel slicer. Six slices were combined to one sample and subjected to in-gel reduction with dithiothreitol, alkylation with iodoacetamide and digestion with trypsin (Thermo, TPCK treated). Nanoflow LC-MS/MS was performed on an EASY-nLC 1200 system (Thermo) coupled to a Fusion Lumos Tribrid Orbitrap mass spectrometer (Thermo), operating in positive mode and equipped with a nanospray source. Peptide mixtures were trapped on an Acclaim PepMap™ 100 C18 trap column (Thermo, column dimensions 2 cm × 100 μm, 5 μm particles) at a flow rate of 1 μl/min. Peptide separation was performed on ReproSil C18 reversed phase column (Dr Maisch GmbH; column dimensions 25 cm × 75 μm, 2.4 μm particles, packed in-house) using a gradient from 0 to 100% B (A = 0.1% FA; B = 80% (v/v) acetonitrile, 0.1% FA) in 120 min and at a constant flow rate of 250 nl/min. The column eluent was directly sprayed into the ESI source of the mass spectrometer. The Orbitrap resolution in MS1 mode was set to 120,000 in profile mode at AGC 4E5 and at m/z range 375–1400 m/z. Fragmentation of precursors was performed in data-dependent mode by HCD at a cycle time of 2 s with a precursor window of 1.6 m/z and a normalized collision energy of 28%; MS2 spectra were recorded in the Orbitrap in centroid mode at a resolution of 30,000 at AGC 5E4. Singly charged precursors were excluded from fragmentation. Dynamic exclusion was set to 60 s and the intensity threshold was set to 3E4. For the analysis, a single LC-MS/MS run was performed for all samples. Raw mass spectrometry data were analyzed using Proteome Discoverer 2.5 (Thermo) including the label free quantitation (Minora) module.

**Statistics and reproducibility**. Prism GraphPad was used to calculate statistical differences. For the survival experiments in Fig. 4e, f a Log-rank (Mantel-Cox) test was used and for statistical significance between groups one-way ANOVA followed by post

hoc analysis by Bonferroni's test was used. All results were confirmed by performing independent replicate experiments, as indicated in each figure legend.

**Reporting summary**. Further information on research design is available in the Nature Research Reporting Summary linked to this article.

## Data availability
Proteomics data have been deposited to the ProteomeXchange Consortium via the PRIDE partner repository with the dataset identifier PXD029008. Source data are provided with this paper (Supplementary Data 2). Any other data are available from the corresponding author upon reasonable request.

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

## Acknowledgements

We thank Dr. A. Theil for advice, Dr. G. Jansen for use of his injection microscope, Dr. Heng-Chi Lee and Dr. Andrew Fire for strains and plasmids and Dr. G. van Cappellen and the Erasmus MC Optical Imaging Center for microscope support. Some strains were provided by the Caenorhabditis Genetics Center (funded by NIH Office of Research Infrastructure Programs P40 OD010440) and the National Bioresource Project for the nematode. This work was supported by the Netherlands Organization for Scientific Research (711.018.007 and ALWOP.494), the Marie Curie Initial Training Network "aDDRess" funded by the European Commission 7th Framework Programme (316390), the European Research Council (advanced grant 340988-ERC-ID), and the gravitation program Cancer-GenomiCs.nl from the Netherlands Organization for Scientific Research. Oncode Institute is partly financed by the Dutch Cancer Society.

## Author contributions

K.L.T., M.v.d.W., C.D., D.H.W.D., M.S., J.A.A. and H.L. performed experiments and analyzed data. K.L.T., M.v.d.W., W.V. and H.L. designed experiments. M.v.d.W., W.V. and H.L. wrote the manuscript. All authors reviewed the manuscript.

## Competing interests

The authors declare no competing interests.
