## [Transparent Peer Review File · Communications Biology]

Reviewers' comments:

Reviewer #1 (Remarks to the Author):

The manuscript by Thijssen et al entitled "C. elegans TFIIH subunit GTF-2H5/TTDA is a non-essential transcription factor indispensable for DNA repair" reports the functional characterization of the C. elegans TTDA ortholog GTF-2H5. The authors established that deficiency of C. elegans TTDA ortholog GTF-2H5 is compatible with life but promotes TFIIH stability in multiple tissues. They also showed that GTF-2H5 is required for nucleotide excision repair and is required for transcription if this process is challenged or compromised.

The major claims of the manuscript are perfectly in line with existing data on TTDA and on the unifying idea that TTD is caused by instability of proteins involved in the regulation of gene expression. The novelty of the work lies in the use of the C. Elegans model which allows to investigate TTDA/GTF2H5 in vivo and provides an advantageous multicellular animal model for studying genetics and pathogenesis of trichothiodystrophy.

The work is convincing and I feel that the C. Elegans model developed in the present manuscript which allows to interrogate the transcription and DNA repair machinery in vivo and will stimulate a large number of experiments.

Below a few suggestions/comments that I hope could clarify/improve the manuscript.

Point 1: Analysis of the gtf-2H5 mutant is based on RT-PCR analysis but no evidence is provided at the protein level (lines 66-81). Did the authors try Western Blot or mass-spectrometry-based approaches to evaluate expression of the tm6360 mutant? Along the same lines to validate the model one should optimally compare expression of GTF-2H5 in non modified animals and in the AID-GFP tagged KI strain (GTF-2H5::AG).

Point 2: To confirm that gtf-2H5 mutants were viable and to exclude that the weakly expressed protein was still partially functional, the authors compared the consequences of auxin-induced depletion of GTF-2H5 and of GFT-2H1, another TFIIH-core subunit and concluded that GFT-2H5 is not essential to embryogenesis (lines 82-103). A full KO approach as attempted in previous work in mice (Theil et al., 2013) would consolidate the findings. Was it attempted?

Point 3: How does the tm6360 mutation compares to the mutations identified in TTD-A patients and in particular to those leading to the synthesis of a truncated polypeptide (Giglia-Mari 2004, Morikawi, 2014)?

Point 4: Along the same line, it could be of interest to characterize the impact of the tm6360 mutation on TFIIH composition using biochemical/proteomic approaches (Pull down/MS for example).

Point 5: Two kinetic pools of the human TTDA protein were identified using a cellular model (Giglia-Marie et al.2006, ref 52). Is this also the case in C. elegans? Does GTF-2H5 exhibit tissue specific mobility as recently it was recently established in the case of the NER endonuclease XPF-1 (Sabatella et al., 2021).

Did the authors try the small molecules targeting the p8/TTD-A dimerization interface (Gervais et al., 2028)?

Point 6: Experiments reported here establish that GTF-2H5 is required for transcription of class II genes when GTF-2H5 KO is combined with loss of GTF-2E1. It could be interesting to investigate consequences of GTF-2H5 deletion on RNA Pol I transcription using a similar strategy.

Response to the reviewer's comments:

Reviewer #1

The manuscript by Thijssen et al entitled "C. elegans TFIIH subunit GTF - 2H5/TTDA is a non - essential transcription factor indispensable for DNA repair" reports the functional characterization of the of C. elegans TTDA ortholog GTF - 2H5. The authors established that deficiency of C. elegans TTDA ortholog GTF - 2H5 is compatible with life but promotes TFIIH stability in multiple tissues. They also showed that GTF-2H5 is required for nucleotide excision repair and is required for transcription if this process is challenged or compromised.

The major claims of the manuscript are perfectly in line with existing data on TTDA and on the unifying idea that TTD is caused by instability of proteins involved in the regulation of gene expression. The novelty of the work lies in the use of the C. Elegans model which allows to investigate TTDA/GTF2H5 in vivo and provides an advantageous multicellular animal model for studying genetics and pathogenesis of trichothiodystrophy.

The work is convincing and I feel that the C. Elegans model developed in the present manuscript which allows to interrogate the transcription and DNA repair machinery in vivo and will stimulate a large number of experiments.

We thank the reviewer for his/her very positive evaluation and high appreciation of our manuscript, highlighting the value of using *C. elegans* as model. Below we address each of the suggestions/comments raised.

Below a few suggestions/comments that I hope could clarify/improve the manuscript.

Point 1: Analysis of the gtf-2H5 mutant is based on RT-PCR analysis but no evidence is provided at the protein level (lines 66-81). Did the authors try Western Blot or mass-spectrometry-based approaches to evaluate expression of the tm6360 mutant? Along the same lines to validate the model one should optimally compare expression of GTF-2H5 in non modified animals and in the AID-GFP tagged KI strain (GTF-2H5::AG).

The reviewer is right that we did not analyze the *gtf-2H5* mutant at the protein level, for which the main reason is the lack of good (*C. elegans* specific) antibodies. However, as suggested, we now performed mass spectrometry to analyze the composition of the TFIIH complex in wild type and *gtf-2H5* mutants, after immunoprecipitating AG::GTF-2H1/p62 (in response to point 4 below). This new data is represented in Figure 3b of the revised manuscript and included as supplementary data 1. It is described in lines 135-147. Importantly, this shows that in three replicate mass spectrometry experiments the mutated *tm6360* protein is not detected (as part of the TFIIH complex) in *gtf-2H5* mutant animals.

Furthermore, due to lack of antibodies, we cannot compare expression of GTF-2H5 in the AID-GFP tagged KI strain with that in non-modified animals. However, we do show that AID-GFP-tagged GTF-2H5 is fully functional in NER (supplementary figure 3A), suggesting that it is expressed at sufficiently high levels. Also, as part of a different project, we performed immunoprecipitation and mass spectrometry on AID-GFP-tagged GTF-2H5, which shows that the tagged protein is normally

incorporated into the TFIIH complex. These data are, however, part of a follow-up project currently running in the lab, which is why we choose not to include these in the current manuscript.

Point 2: To confirm that *gtf-2H5* mutants were viable and to exclude that the weakly expressed protein was still partially functional, the authors compared the consequences of auxin-induced depletion of GTF-2H5 and of GTF-2H1, another TFIIH-core subunit and concluded that GTF-2H5 is not essential to embryogenesis (lines 82-103). A full KO approach as attempted in previous work in mice (Theil et al., 2013) would consolidate the findings. Was it attempted?

We did not attempt a full knockout approach, because we observe full depletion of GTF-2H5 after activating TIR1 by adding auxin (shown in Supplementary Fig. 3b). Therefore, we consider this auxin-depletion functionally similar as a full KO. In spite of this full GTF-2H5 depletion, we never observed any effect on viability even after observing hundreds of animals (depicted in Figure 2e). Moreover, in mice and humans full TTDA knockout renders cells equally UV sensitive as XPA deficiency, which is what we also observed when comparing the *gtf-2H5* mutants with *xpa-1* mutants (Fig 4a). In contrast, partially functional mutant TTD in human cells leads to only partial UV-sensitivity (Theil et al, 2013). This confirms that the *gtf-2H5(tm6360)* functionally behaves as a KO. In addition, with our new mass spectrometry analysis (Figure 3b), we now show that there is no detectable incorporation of (mutant) GTF-2H5 in the TFIIH complex in *gtf-2H5* mutants. Therefore, we are confident that the *gtf-2h5(tm6360)* functionally represents a full KO.

Point 3: How does the *tm6360* mutation compares to the mutations identified in TTD-A patients and in particular to those leading to the synthesis of a truncated polypeptide (Giglia-Mari 2004, Morikawi, 2014)?

We thank the reviewer for this interesting question, which we now address in the manuscript in lines 84-91. In Supplementary Fig. 1d we show a comparison of the known human TTDA mutations with the *tm6360* mutation in *C. elegans*. As we now discuss in the revised manuscript, this comparison suggests that *tm6360* will not lead to expression of a partial functional protein as is the case in human TTD-A patients.

Point 4: Along the same line, it could be of interest to characterize the impact of the *tm6360* mutation on TFIIH composition using biochemical/proteomic approaches (Pull down/MS for example).

We agree with the reviewer that characterizing TFIIH composition after loss of GTF-2H5 is interesting. Therefore, using AG::GTF-2H1 immunoprecipitation and mass spectrometry analysis, we now include in the revised manuscript a label-free quantification comparison of TFIIH complex composition between wild type and mutant animals, shown in Figure 3b and supplementary data 1. Interestingly, this confirms the lowered levels of TFIIH in *gtf-2H5* mutants (supplementary data 1), but indicates that the complex itself, apart from GTF-2H5, is intact (Figure 3b). We also observed binding of some NER proteins to TFIIH (without DNA damage), which is probably because with this sensitive proteomics procedure we rapidly pick up any protein with affinity for TFIIH. Interestingly, in contrast to wild type animals, XPA-1 was not found to associate with TFIIH in *gtf-2H5* mutants, in line with the NER defect of these mutants.

Point 5: Two kinetic pools of the human TTDA protein were identified using a cellular model (Giglia-Marie et al. 2006, ref 52). Is this also the case in *C. elegans*? Does GTF-2H5 exhibit tissue specific mobility as recently it was recently established in the case of the NER endonuclease XPF-1 (Sabatella et al., 2021).

We have not observed clear fluorescence of GTF-2H5::AG outside of the nucleus in *C. elegans* tissues, which is why we cannot confirm whether two kinetic pools exist *in vivo* in this model organism. However, endogenous expression of GTF-2H5 is relatively low (compared to transgenic overexpression) and *C. elegans* tissues typically show high background fluorescence levels (see e.g. supplementary Fig 2), it is difficult to distinguish whether low GTF-2H5::AG fluorescence may be present in the cytosol. For this reason, we have not commented on the possibility of two kinetic pools or cytosolic expression in *C. elegans* cells. It is for sure interesting to perform mobility studies on TFIID in different tissues of *C. elegans*, as we have done for XPF-1, and compare these to similar studies done in human cells and mouse tissue. However, these studies are beyond the scope of this manuscript and are part of a new, ongoing project in the lab.

Did the authors try the small molecules targeting the p8/TTD-A dimerization interface (Gervais et al., 2028)?

This is an interesting suggestion which we have not tried. The difficulty with using chemical inhibitors in *C. elegans* is that it is unknown whether and how these will be taken up by the animal and thus which exact incubation conditions should be used. Incubation conditions should therefore be experimentally tested and optimized, which would take considerable time and effort. As for this manuscript there is no specific research question that we would address using these compounds in *C. elegans*, we have not tried these inhibitors yet.

Point 6: Experiments reported here establish that GTF-2H5 is required for transcription of class II genes when GTF-2H5 KO is combined with loss of GTF-2E1. It could be interesting to investigate consequences of GTF-2H5 deletion on RNA Pol I transcription using a similar strategy.

We agree that it would be interesting to study the interplay between GTF-2H5 deficiency and RNA Pol I transcription, as TFIID has been implicated in RNA pol I transcription. However, we think that this would be beyond the scope of our current manuscript and would entail a whole new project. This is because TFIID promotes RNA pol I transcription elongation and not initiation as with RNA pol II. Thus, we would have to investigate which transcription factors (other than GTF-2E1) we could downregulate to obtain a synthetic phenotype with GTF-2H5. Furthermore, unfortunately the molecular mechanisms of transcription regulation, in particular that of RNA Pol I, have not been firmly established yet in *C. elegans*. For instance, *rpoa-1* is predicted to be an ortholog of POLR1A, but this has not been functionally verified. Also, clear orthologs of Pol I transcription factors, such as UBF and SL1, have not yet been identified. However, as we think that the suggestion of the reviewer is very interesting for follow-up research, we have added it to the end of the manuscript text.

REVIEWERS' COMMENTS:

Reviewer #1 (Remarks to the Author):

Point-by-point responses are convincing and the revised version of the manuscript (COMMSBIO-21-1549A) addresses key concerns identified in the first version of the article; In particular, the new mass spectrometry experiments in wild-type and mutant worms provide key information on gtf-2H5 at the protein level.